# Microbiome: The Next Frontier in Psychedelic Renaissance

Robert B. Kargbo [ID]

Usona Institute, 2800 Woods Hollow Rd., Madison, WI 53711-5300, USA; kargborb@gmail.com

**Abstract:** The psychedelic renaissance has reignited interest in the therapeutic potential of psychedelics for mental health and well-being. An emerging area of interest is the potential modulation of psychedelic effects by the gut microbiome—the ecosystem of microorganisms in our digestive tract. This review explores the intersection of the gut microbiome and psychedelic therapy, underlining potential implications for personalized medicine and mental health. We delve into the current understanding of the gut–brain axis, its influence on mood, cognition, and behavior, and how the microbiome may affect the metabolism and bioavailability of psychedelic substances. We also discuss the role of microbiome variations in shaping individual responses to psychedelics, along with potential risks and benefits. Moreover, we consider the prospect of microbiome-targeted interventions as a fresh approach to boost or modulate psychedelic therapy's effectiveness. By integrating insights from the fields of psychopharmacology, microbiology, and neuroscience, our objective is to advance knowledge about the intricate relationship between the microbiome and psychedelic substances, thereby paving the way for novel strategies to optimize mental health outcomes amid the ongoing psychedelic renaissance.

**Keywords:** psychedelic renaissance; mental health; therapeutic potential; gut microbiome; psychedelic substances; personalized medicine; gut–brain axis; mood regulation; microbiome-targeted interventions; psychopharmacology



## 1. Introduction

The paradigm of mental health treatment is undergoing a significant shift characterized by a resurgent interest in the therapeutic potential of psychedelic substances [1–3]. Concurrently, the role of the gut microbiome in mental health has emerged as a prominent field of study [4,5]. The intersection of these two fields has given rise to an exciting frontier of research in the psychedelic renaissance, investigating the interaction between the gut microbiome and psychedelics. This review aims to explore the state of knowledge in this burgeoning field [6], focusing on understanding the interplay between the gut microbiome and the effects of psychedelic substances, and how this interaction may shape the future of personalized mental health treatment [7–9].

The role of the microbiome in mental health is based on the intricate relationship between the gut and the brain, often referred to as the gut–brain axis [10,11]. The gut–brain axis is a complex bidirectional communication system that integrates neural, hormonal, and immunological signaling between the gut and the brain [12–14]. It has been implicated in a variety of psychological and neurological conditions, including anxiety, depression, stress, cognitive impairment, and sleep disorders, and plays a vital role in mental health.

A central player in this communication system is the gut microbiome, a richly diverse ecosystem of microorganisms in the human gastrointestinal tract [9,15]. The microbiome contributes significantly to our overall health, influencing digestion, immunity, mood, and cognition. Over the past decade, research has unveiled the profound influence that the gut microbiome can have on the brain and behavior, leading to the emergence of the field of psychobiotics, which explores how modifications of the microbiome can affect mental health.

The gut microbiome's role in modulating the effects of drugs has been well-established in the context of various medications, including antipsychotics and antidepressants [16,17]. More recently, research has turned its attention to the potential role of the gut microbiome in modulating the effects of psychedelic substances. The therapeutic efficacy of psychedelics has gained renewed interest, driven by promising results from clinical trials investigating their potential to treat mental health disorders such as depression, anxiety, and post-traumatic stress disorder (PTSD) [18,19].

Evidence suggests that the gut microbiome could be implicated in the metabolism and bioavailability of psychedelic substances and their therapeutic effects. For example, recent studies have indicated that specific gut bacteria can modulate the metabolism of *N,N*-dimethyltryptamine (DMT), a psychoactive compound found in ayahuasca, which may influence the bioavailability and pharmacological effects of DMT in the host [19,20]. Furthermore, the individual variability in gut microbiome composition may influence the bioavailability and effects of psychedelic substances, emphasizing the potential for a personalized approach in psychedelic therapy. In Figure 1, Misera et al. illustrate how the gut microbiota influences psychiatric treatment efficacy [4]. Antipsychotic drugs alter the microbiota composition, which can mitigate psychiatric symptoms and potentially induce metabolic disorders—a common reason for treatment discontinuation. Probiotic supplementation may alleviate metabolic issues and augment drug effectiveness, highlighting a complex interplay between microbiota, psychopharmacology, and mental health outcomes.

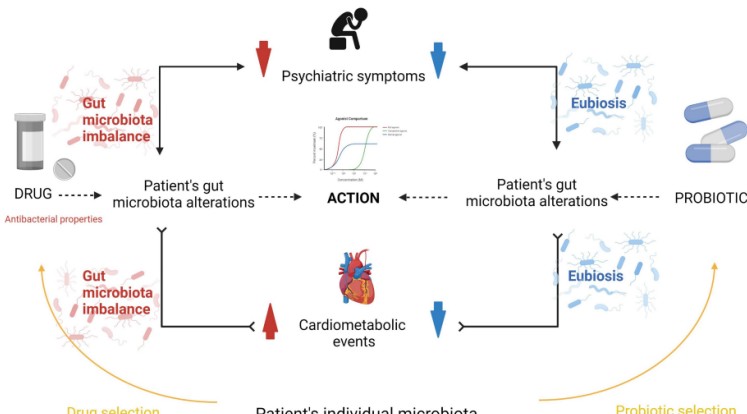

**Figure 1.** The influence of microbiota on the effectiveness of treatment in psychiatry. (Reprinted with permission from Ref. [4]. Copyright 2012 Frontiers Journals.).

However, our understanding of this intricate interplay between the microbiome and psychedelic substances is in its infancy. There is much we still do not know, including the extent to which the gut microbiome modulates the metabolism and effects of different psychedelic substances, how alterations in the gut microbiome might contribute to individual variability in response to these substances, and the potential risks and benefits of microbiome-mediated effects of psychedelics. This mini-review aims to examine the current evidence, shed light on these questions, and set the direction for future research in this exciting new field of psychedelic science.

## 2. Understanding the Gut–Brain Axis

The gut–brain axis represents a complex bidirectional communication system that establishes a connection between the central and enteric nervous systems, playing an indispensable role in maintaining homeostasis [21,22]. The gut–brain axis encompasses multiple communication channels, including the immune system, the vagus nerve, and the production of various hormones and neurotransmitters [23,24]. Importantly, recent studies have highlighted the crucial role that the gut microbiota plays in modulating this axis [25,26].

The brain can influence the composition of the gut microbiota through the autonomic nervous system (ANS), affecting gut motility and secretion, as well as immune cell function [27–29]. Conversely, the gut microbiota can affect the brain's activity through the production of metabolites, modulation of immune responses, and direct interactions with enteroendocrine cells and the enteric nervous system (ENS) [30,31]. Figure 2 shows the bidirectional communication between the gut and the brain [32]. Afferent and efferent brain neurons connect and signal through multiple pathways, including the ANS, ENS, and hypothalamic–pituitary–adrenal (HPA) axis.

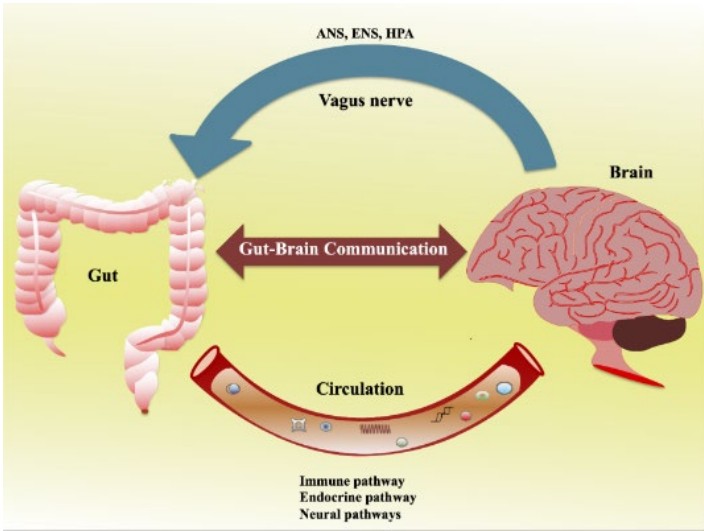

**Figure 2.** Schematic diagram showing the bidirectional communication between the gut and the brain. (Reprinted with permission from Ref. [32]. Copyright 2008 published by MDPI).

The ENS is a complex network that controls intrinsic gut functions like motility, secretion, and absorption. Multiple pathways such as the ANS, ENS, HPA axis, immune, endocrine, and neural pathways influence communication between the gut and brain [33–35]. The ENS communicates with the central nervous system (CNS) via intestinofugal neurons, and sensory information travels via primary afferent neurons. The ANS, consisting of sympathetic and parasympathetic nerves, in conjunction with neuronal and neuroendocrine signaling, controls vital functions and CNS-mediated gut changes. The ANS impacts gut physiology directly via the CNS, whereas gut microbiota communicates through its metabolites and interacts with the ANS gut synapses. Furthermore, the ANS can modulate gut immune responses directly or indirectly via microbial interactions.

One of the most striking examples of the gut microbiota's influence on brain function comes from studies of germ-free mice [36,37]. These animals exhibit various alterations in brain chemistry and behavior compared to conventionally raised mice, suggesting a fundamental role for the gut microbiota in brain development and function.

Certain members of the gut microbiota can produce neurotransmitters and neuromodulators, including gamma-aminobutyric acid (GABA), serotonin, dopamine, and short-chain fatty acids (SCFAs), which can influence brain function both directly and indirectly [38,39]. For instance, SCFAs can modulate the function of microglia, the primary immune cells in the brain, thereby influencing neuroinflammation and neurodegeneration [40]. Furthermore, some gut bacteria can metabolize dietary tryptophan into indole derivatives, which can activate the aryl hydrocarbon receptor (AHR), a ligand-activated transcription factor that regulates immune responses and maintains the integrity of the gut barrier [41,42].

The gut–brain axis also plays a crucial role in stress responses. Exposure to stress can alter the composition of the gut microbiota, a phenomenon termed dysbiosis, leading to increased intestinal permeability, also known as "leaky gut" [43,44]. This can lead to the translocation of bacterial components into the bloodstream, triggering systemic inflammation, which, in turn, can affect brain function and behavior [45,46].

The gut–brain axis is a multifaceted communication system linking the brain with the gut microbiota, playing a fundamental role in health and disease. An improved understanding of the gut–brain axis could pave the way for novel therapeutic strategies in treating various psychiatric and neurological disorders, including those potentially modulated by psychedelic substances.

### 3. Microbiome and Psychedelic Interaction

The human gut microbiome profoundly influences our health and well-being, not only in terms of physical health but also as a modulator of brain function and behavior, including mood, cognition, and stress responses [23,47]. With the ongoing psychedelic renaissance exploring the therapeutic potentials of psychedelic substances like psilocybin, lysergic acid diethylamide (LSD), and ayahuasca, it is essential to consider the role of the gut microbiome in this narrative.

Psychedelic substances can induce potent changes in consciousness, leading to significant alterations in perception, mood, and cognitive processes [48,49]. The profound effects of these substances have sparked renewed interest in their therapeutic potential, particularly for mental health disorders such as depression, anxiety, and PTSD [50,51].

In Figure 3, Kelly et al. illustrate how the microbiota–gut–brain (MGB) axis may modulate responses to psychedelic therapy, acting as a biofeedback system [52]. Initial MGB activity could help identify individuals more likely to benefit from such therapy. Before treatment, adjustments to the MGB axis could enhance responsiveness. During treatment, the MGB axis might affect psychedelic drug metabolism variability, whereas post-treatment, reinforcing MGB signaling could promote and sustain beneficial behavioral changes toward homeostasis.

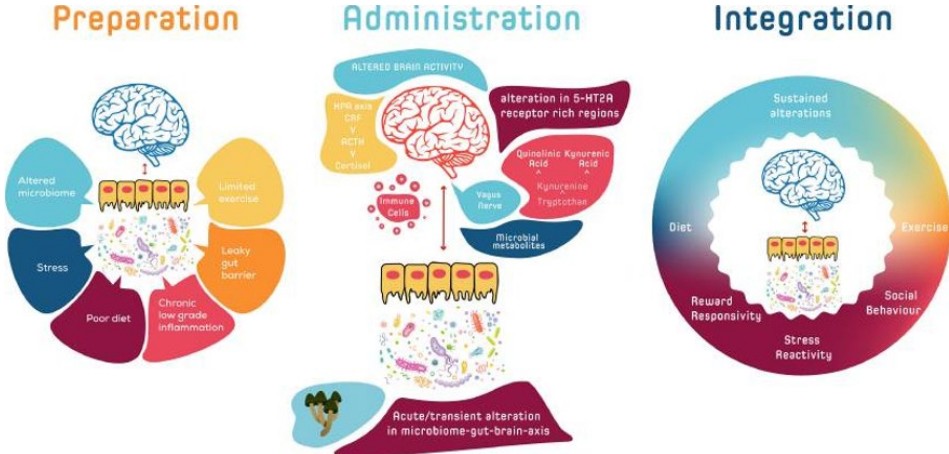

**Figure 3.** Potential bidirectional interactions between the host, microbiota, and psychedelics involving various biofeedback mechanisms. (Reprinted with permission from Ref. [52]. Copyright 2022 published by Elsevier).

Furthermore, recent evidence suggests the gut microbiome might play a role in the metabolism and bioavailability of these psychedelic substances, which could impact their pharmacological effects. For instance, Nichols et al. found that the gut bacterium Bifidobacterium modulates the metabolism of *N,N*-dimethyltryptamine (DMT), a psychoactive compound found in ayahuasca [53]. This finding hints at the possibility that the therapeutic effects of psychedelics may, in part, be contingent upon the composition of an individual's gut microbiota.

The metabolic capacity of the gut microbiota extends to a variety of psychedelic substances, including psilocybin, LSD, and mescaline. Research by Mezquita et al. indicated that certain bacterial strains possess the necessary enzymes to convert psilocybin, found in "magic mushrooms", into its active metabolite psilocin [54]. Furthermore, Nichols et al.

found that the gut bacterium Enterococcus faecalis possesses the enzyme lysergic acid diethylamide *N*-demethylase, which can degrade LSD [55].

Another line of investigation focuses on mescaline, a psychoactive compound found in the peyote cactus. Preliminary research by Saito et al. suggested that specific bacterial strains could metabolize mescaline into different metabolites in vitro [56]. While these studies provide preliminary evidence of microbiome involvement in the metabolism of various psychedelic compounds, further research is necessary to understand the extent and implications of these interactions for the bioavailability, effects, and individual variability in response to these substances.

The gut microbiota modulates the gut–brain axis, a bidirectional communication system between the enteric and central nervous systems, and can influence various brain functions [30]. For example, alterations in the gut microbiota have been linked to mood disorders like depression and anxiety, suggesting a role for the microbiome in modulating emotional responses [57,58]. However, more research is needed to test these hypotheses and understand the precise mechanisms involved.

The potential interaction between the gut microbiome and psychedelic substances represents a promising area of research within the psychedelic renaissance. By investigating these interactions, we might gain novel insights into how psychedelics work, the factors influencing their effects, and new methods to optimize their therapeutic potential.

## 4. Interpersonal Variability in Psychedelic Response

Psychedelic substances, despite their demonstrated therapeutic potential, have long been recognized to exhibit a high degree of interpersonal variability in response [59–61]. Individuals who use the exact dosage of the same substance can experience vastly different subjective effects. Several manifestations, including intensity and duration of impact, subjective experiences, and potential for therapeutic results, can reveal this variability. Recognizing and understanding these differences is crucial, especially when considering the therapeutic applications of psychedelic substances in personalized medicine.

Traditionally, a range of factors account for the variability observed in the psychedelic experience. One of the most recognized is "set and setting" [62–64]. "Set" refers to the mindset, expectation, and psychological state of the individual at the time of taking the psychedelic. Meanwhile, "setting" refers to the physical and social environment where individuals use the substance. This concept suggests that both the individual's internal mental state and the external environment significantly influence the individual's experience with the psychedelic. Psychedelic drugs offer therapeutic potential but can also induce adverse effects, making it crucial to predict individual reactions.

Figure 4 illustrates a systematic review of 14 studies, which found that traits like absorption, openness, acceptance, and a state of surrender correlate with positive experiences, whereas those low in openness and surrender or certain negative psychological conditions are more likely to have adverse reactions [65]. Age, experience with psychedelics, 5-HT2AR binding potential, executive network node diversity, and rACC volume might influence reactions.

Another aspect contributing to interpersonal variability is genetic predisposition [55,66]. It is well established that genetic factors may influence the metabolism of various substances, including psychedelics, and this can potentially affect their pharmacokinetics and pharmacodynamics. For instance, polymorphisms in enzymes involved in the metabolism of psychedelic substances, such as monoamine oxidases and cytochrome P450 enzymes, can lead to differences in the metabolism rate among individuals, thereby affecting the substances' bioavailability and effects [67].

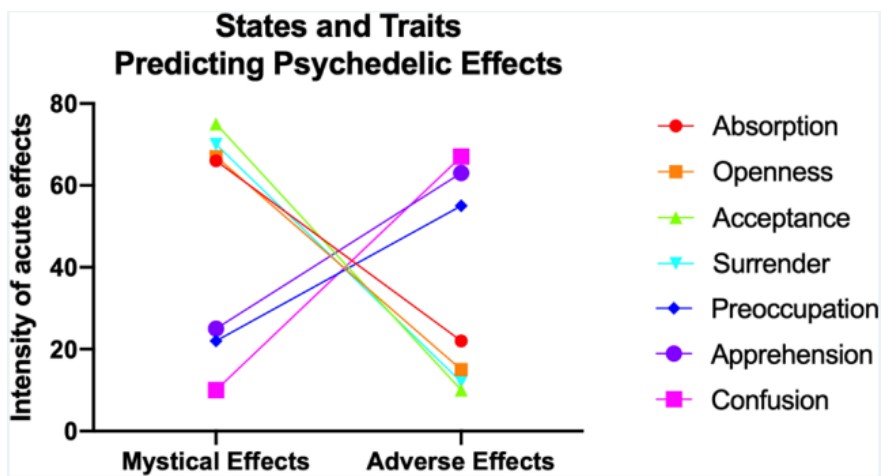

**Figure 4.** Correlation of Individual Traits, Neurobiological Factors, and Psychedelic Experiences: A Systematic Review of 14 Studies. (Reprinted with permission from Ref. [65]. Copyright 2021 American Chemical Society).

The emerging role of the gut microbiome adds another layer of complexity to this equation. As discussed earlier, the gut microbiome can affect the metabolism of psychedelic substances, suggesting it may play a role in mediating the interpersonal variability in psychedelic response. Differences in gut microbiome composition between individuals could lead to variability in the bioavailability and effects of psychedelic substances. For example, if one individual's gut microbiota metabolizes a psychedelic substance more efficiently than another's, this could lead to differences in the effects experienced by the two individuals, even if they consume the same dose [54].

Additionally, the gut microbiome's role in modulating the gut–brain axis could influence an individual's psychological response to psychedelics. It is plausible that an individual's gut microbiota could affect their mood and cognition, which could influence their response to psychedelic substances [57]. Moreover, recent research suggests that the gut microbiota can affect the brain's serotonergic system, a primary target of many psychedelic substances [30].

While we are just beginning to understand how the gut microbiome might influence the interpersonal variability in psychedelic response, this area of research has profound implications. Understanding the factors contributing to this variability can potentially lead to the optimization of psychedelic therapy to meet individual needs. This process might involve individual-specific customization of psychedelic dosage or type, based on parameters like genetic makeup, gut microbiome composition, or psychological disposition.

The variability in the psychedelic response among individuals is a complex phenomenon influenced by numerous factors, including mindset, environment, genetics, and, potentially, the gut microbiome. More research is needed to understand these factors and their interactions, but the potential payoff is substantial: a more personalized and practical approach to psychedelic therapy.

## 5. Implications for Mental Health Treatment

Psychedelics, once stigmatized and marginalized in the medical community, are now in the spotlight as a potentially transformative treatment for various mental health disorders [68]. The recent rediscovery of the therapeutic potential of these substances, sometimes referred to as the "psychedelic renaissance", has offered new hope for patients suffering from treatment-resistant mental health conditions such as depression, anxiety, PTSD, and addiction [69,70].

The understanding that the gut microbiome may influence the response to psychedelic substances brings about a new frontier in applying these substances for mental health treatment. As mentioned, variability in psychedelic experiences is significant, and this

individuality of response could be partially modulated by the unique microbial composition of each person's gut [71]. If further research confirms this hypothesis, we could soon be at the cusp of tailoring psychedelic therapy to the individual's microbiome, thus making treatment more effective and personalized.

Despite strict regulations, the US has the most ongoing clinical trials on psychedelics (Figure 5), hinting at possible future easing of these laws. Switzerland, more accepting of psychedelics, leads in per-capita studies, reflecting the influence of cultural acceptance on research [72]. The predominance of US trials in this study may be due to a bias from using the Clinicaltrials.gov database, whereas other databases exist. Reviewing clinical trials over time reveals an evolution in hypotheses, with initial studies before 2010 primarily focused on using psychedelics to improve mood in patients undergoing other treatments. More recently, psychedelics have been used as the primary treatment for various diseases, particularly psychiatric ones. The methodological robustness of these studies has also improved over time, with more trials using quadruple masking.

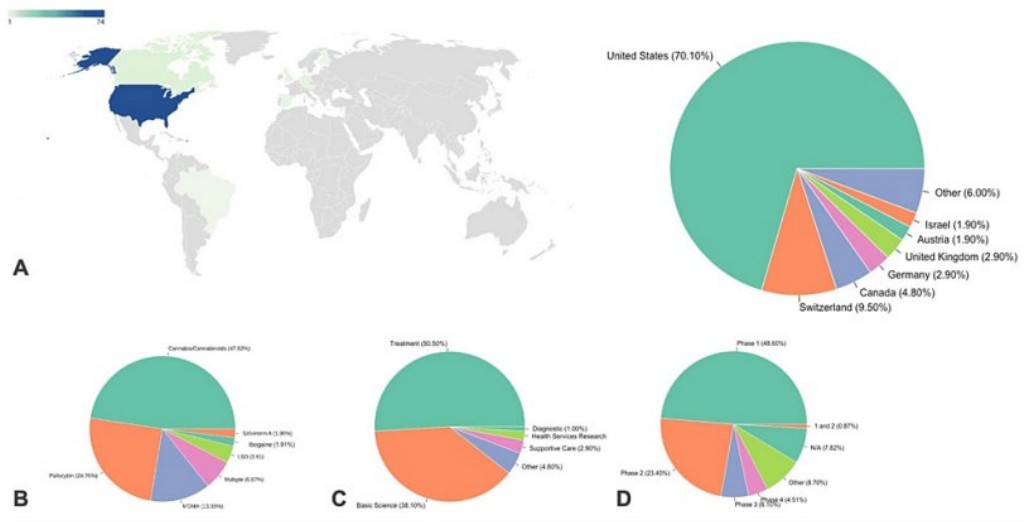

**Figure 5.** Overview of Clinical Trials: Nationality, Analyzed Psychedelic Drugs, Trial Type, and Current Stages. (**A**) Clinical trials by nationality. (**B**) Psychedelic drugs under analysis in each trial. (**C**) Type of clinical trial. (**D**) Stage of currently reported clinical trials underway. (Reprinted with permission from Ref. [72] Copyright published by Springer Nature).

Moreover, if the gut microbiome can metabolize psychedelic substances and consequently affect their bioavailability, it may be possible to modulate the gut microbiome to enhance the bioavailability of these substances [54]. This process could result in more efficient and potent treatments, minimizing the required dosage amount of the compound and potentially reducing side effects.

Lastly, considering the emerging concept of a "psychedelic diet" or pre- and probiotic supplementation prior to psychedelic therapy, it might be possible to prepare the gut microbiome to maximize the therapeutic effects of psychedelics [73,74]. However, further research is needed to validate the efficacy of such approaches.

The interactions between the gut microbiome and psychedelic substances could potentially revolutionize how we approach mental health treatment. While more research is needed to fully understand these interactions, the possibility of a more personalized and effective mental health treatment paradigm is undoubtedly on the horizon.

## 6. Microbiome-Targeted Interventions in Psychedelic Therapy

The symbiotic relationship between humans and the trillions of microbial organisms residing in the gut, collectively known as the gut microbiome, has received increased scientific interest over the past decade.

Psychedelic substances, such as psilocybin, LSD, and 3,4-Methylenedioxymethamphetamine (MDMA), have shown promising results in the treatment of mental health disorders, including depression, anxiety, and PTSD [69,70]. Yet, the therapeutic effects of these substances vary considerably among individuals, which may be due to differences in the gut microbiome composition [71].

Psychedelic substances are exogenous compounds or xenobiotics, which can be metabolized by the gut microbiota, potentially affecting their bioavailability and therapeutic effects. Therefore, microbiome-targeted interventions aimed at manipulating the gut microbiome could theoretically modulate the therapeutic efficacy of psychedelics.

Several potential microbiome-targeted interventions include prebiotics, probiotics, and fecal microbiota transplantation (FMT). Prebiotics are dietary substances that promote the growth of beneficial gut bacteria, enhancing overall gut health. Probiotics are live bacteria that confer a health benefit to the host when consumed in adequate amounts. Both prebiotics and probiotics could potentially influence the metabolism and effects of psychedelic substances by altering the gut microbiome composition [75,76].

MGB communication is studied by manipulating the microbiota, using methods such as germ-free (GF) animal models, microbiome depletion with antibiotics, and FMT (Figure 6) [77]. GF models reveal the microbiome's role in stress response, anxiety, social behavior, and cognition. The transfer of microbiota through FMT potentially carries the risk of transmitting disorders such as depression while also demonstrating therapeutic potential in treatments for gastrointestinal and psychiatric conditions. Phage therapy, involving viruses that infect specific bacteria, offers the potential in modulate microbiome composition, although its use is currently limited to research. "Postbiotics", nonviable bacterial products or metabolites, notably short-chain fatty acids, also play a significant role in the host, with their production encouraged by high-fiber diets. These methods could be employed to target the MGB axis in psychedelic therapy.

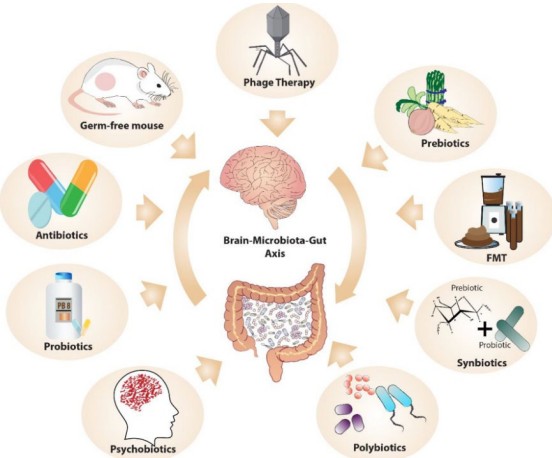

**Figure 6.** Methods of Manipulating Microbiota–Gut–Brain Axis in Psychedelic Therapy Research. (Reprinted with permission from Ref. [77]. Copyright published by SAGE Publications).

FMT, the process of transferring fecal bacteria from a healthy donor to a recipient, represents another potential method to alter the gut microbiome and, possibly, the response to psychedelic substances. It has been successfully used in the treatment of Clostridium difficile infections and is currently explored as a treatment for other disorders associated with gut microbiota imbalances, such as irritable bowel syndrome (IBS) and inflammatory bowel disease (IBD) [78].

There is a growing interest in the gut microbiome's role in modulating the effects of psychedelic therapy. Future research should focus on understanding the complex interactions between the gut microbiome and psychedelic substances and explore the potential of microbiome-targeted interventions to enhance the therapeutic efficacy of psychedelics.

## 7. Personalized Medicine and Psychedelics

The current era of medicine is witnessing a shift from the conventional one-size-fits-all treatment approaches to more personalized, patient-specific strategies. This transition is fueled by advancements in genomics, proteomics, and other molecular diagnostic tools that enable the identification of individual-specific disease risk factors, prognostic indicators, and therapeutic targets. Despite their historical, socio-cultural roots, psychedelic therapies are no exception to this trend.

Personalized psychedelic therapy refers to tailoring psychedelic treatment to meet an individual's unique physiological, psychological, and experiential needs. The consideration for such customization ranges from genetic and metabolic differences that influence the pharmacokinetics of psychedelic substances to distinct psychosocial contexts that can shape an individual's subjective psychedelic experience and subsequent therapeutic outcomes.

Genetic variations play a critical role in the metabolism and effects of psychedelics. Cytochrome P450 enzymes, primarily CYP2D6, metabolize many psychedelics, including psilocybin and DMT. Genetic polymorphisms in the CYP2D6 gene are associated with variations in enzyme activity, leading to significant interindividual variability in drug metabolism. For instance, poor metabolizers can experience intensified and prolonged psychedelic effects due to slow drug metabolism, whereas ultra-rapid metabolizers may require higher doses for therapeutic efficacy [79].

Furthermore, due to genetic polymorphisms, there is considerable variability in the 5-HT2A receptor—the primary target of psychedelics. Differences in receptor density, signaling efficacy, and downstream effects may lead to differential responses to psychedelics [80]. Research actively links polymorphisms of the HTR2A gene with susceptibility to psychological disorders and the response to psychedelic treatment [81].

Psychedelic experiences are deeply personal and highly influenced by the individual's mindset and environment—a concept known as "set and setting." Individual personality traits, mental states, expectations, cultural backgrounds, and physical and social environment can significantly modulate psychedelic therapy's subjective experience and therapeutic outcomes. Recognizing these factors and tailoring the set and setting to the individual's needs can enhance the safety and efficacy of psychedelic therapy [82].

Personalized psychiatry faces challenges due to the reliance on categorical diagnostic systems and small-scale studies, which fail to capture mental health complexity. Though useful in clinical settings, they often overlap and do not provide specific biological markers of mental health. Small effect sizes in mental health studies necessitate integration of multiple variables for accurate prediction models. A shift toward a transdiagnostic, dimensional approach, deconstructing diagnoses into dimensional constructs, may enhance treatment precision. The RDoC (Research Domain Criteria) neuroscientific framework integrates developmental processes and environmental inputs, aiming to identify specific biosignatures for better mental health outcomes (Figure 7) [83]. This approach could revolutionize the precision of psychedelic therapy.

**Figure 7.** Integration of Transdiagnostic and Dimensional Approach in Personalized Psychiatry for Enhanced Treatment Precision. (Reprinted with permission from Ref. [83] Copyright 2012 Frontiers Journals).

As discussed earlier, the microbiome–gut–brain axis adds another level of complexity to personalized psychedelic therapy. The gut microbiome's composition and function, influenced by diet, lifestyle, antibiotics use, and other factors can affect the metabolism and neuropharmacology of psychedelics. Personalizing psychedelic therapy might involve modulating the gut microbiome to optimize psychedelic effects [84].

Integration, the process of making sense and deriving insights from the psychedelic experience, is a crucial part of psychedelic therapy [85,86]. The approach is highly individualized, depending on the person's cognitive styles, emotional processing, cultural context, and personal narratives. Ensuring the integration process aligns with the individual's unique needs can improve therapeutic outcomes [87].

Despite its potential, personalized psychedelic therapy poses several challenges. It requires an interdisciplinary approach integrating psychopharmacology, genetics, microbiology, psychology, and other fields. In-depth knowledge of the individual patient is required, which needs thorough pre-treatment assessments. Also, while some personalization aspects, such as set and setting, can be addressed relatively quickly, others, such as genetic testing and microbiome modulation, require more resources and advanced technologies. Further research is needed to understand the best practices for personalizing psychedelic therapy and to ensure that these personalized approaches are accessible and equitable [88,89].

Personalized psychedelic therapy represents a promising frontier in the psychedelic renaissance. It capitalizes on the unique characteristics of psychedelic therapy—its physiological effects, subjective experiences, and potential for profound personal insights. By considering the individual special needs and contexts of each patient, personalized psychedelic therapy has the potential to maximize therapeutic benefits, minimize risks, and contribute to the evolving understanding of psychedelic medicines.

## 8. Limitations and Future Research Directions

While the renaissance of psychedelic research is undeniably exciting, yielding promising results in treating various mental health disorders, it is essential to remain aware of the current limitations and challenges in this field. Recognizing these challenges can help inform future research directions, ensuring the responsible and effective development and application of psychedelic therapies.

One primary limitation is the heterogeneity of the psychedelic experience, which can lead to inconsistent outcomes in clinical trials. The psychedelic experience is influenced by numerous factors, including the individual's mindset, the environment ("set and setting"), and the therapeutic relationship [82]. Thus, standardizing these aspects across diverse patient populations and treatment settings poses a significant challenge. Future research should aim to identify reliable and feasible methods for controlling these factors in clinical trials to achieve more consistent results.

Furthermore, the pharmacokinetics of psychedelics varies significantly among individuals, partly due to genetic polymorphisms affecting drug metabolism and target receptor function [79,90]. This variability can affect the safety and efficacy of psychedelic therapies, requiring dose adjustments or alternative treatment options for specific individuals. Future research could explore pharmacogenomic approaches to personalize psychedelic therapy while investigating the safety and efficacy of different dosing regimens.

Figure 8 illustrates the brain–gut–microbiota axis, allowing bidirectional communication between the gut and CNS via complex, poorly understood mechanisms like neural, endocrine, immune, and metabolic pathways. Gut microbes can produce most brain neurotransmitters, influencing the CNS via multiple mechanisms. For example, probiotic Bifidobacteria can increase the serotonin precursor tryptophan. Lactobacilli species can modify GABA metabolism, changing brain GABA receptor expression and behavior. Gut–brain interaction also stimulates the hypothalamic–pituitary–adrenal (HPA) axis, inducing cortisol secretion, the body's primary stressor system. Environmental factors and psychological or physical stress can affect this system, subsequently impacting gut

microbiota/barrier function (Figure 8) [91]. To understand and exploit these mechanisms in clinical settings fully, further research is required.

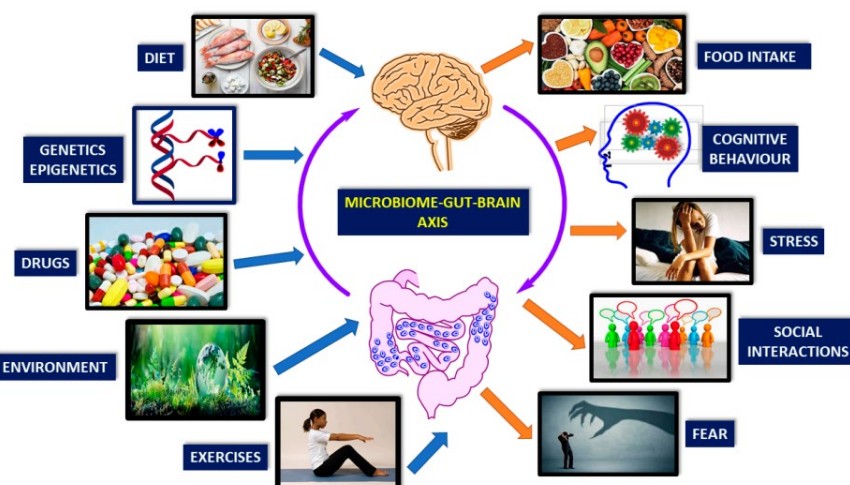

**Figure 8.** Limitations in Current Understanding and Potential Directions for Future Research in the Brain–Gut–Microbiota Axis. (Reprinted with permission from Ref. [91]. Copyright 2008 published by MDPI).

Another critical limitation is the limited understanding of the long-term effects of psychedelic therapy. While short-term therapeutic effects are well-documented, more research is needed to assess the duration of these effects and potential long-term risks [92,93]. Long-term follow-up studies are essential to determine the sustainability of therapeutic outcomes and identify any late-onset adverse effects [94].

Additionally, integrating psychedelic experiences into one's daily life is a vital but often overlooked aspect of psychedelic therapy. Integration involves integrating the psychedelic experience and applying any insights gained to promote positive changes in one's life. The lack of standardized guidelines and professional training for integration therapy is a significant challenge that future research needs to address [95,96].

There are also legal and regulatory challenges to psychedelic research. Despite recent changes in some regions, the use of psychedelics remains heavily restricted in many countries, hindering research and the accessibility of psychedelic therapy [5]. Continued advocacy for policy reform, backed by rigorous scientific evidence, is needed to reduce these barriers.

Looking at the microbiome's influence on psychedelic therapy, a novel but rapidly evolving area of research, there is still much to learn. Existing evidence points to a potential role of gut microbiota in modulating the effects of psychedelics, suggesting the potential of microbiome-targeted interventions to enhance psychedelic therapy. However, our understanding of the intricate relationships between the gut microbiota, the brain, and psychedelic substances still needs to be improved. Future research should focus on elucidating these mechanisms and conducting controlled trials of microbiome-targeted interventions in conjunction with psychedelic therapy [55].

Furthermore, there is a need for research into methods for mitigating potential adverse effects of psychedelic therapy, such as the development of persistent psychosis or hallucinogen-persisting perception disorder (HPPD) in susceptible individuals [97].

Finally, an important future direction for psychedelic research is to continue exploring the therapeutic potential of these substances for a broader range of disorders. While most research has focused on mental health disorders, emerging evidence suggests potential applications in neurology, immunology, and other medical fields [20].

### 9. Summary and Conclusions

Psychedelics have gained substantial attention in recent years for their potential therapeutic applications in treating mental health disorders, particularly depression, anxiety, and PTSD. This renewed interest, often referred to as the psychedelic renaissance, is rooted in acknowledging the unique capacity of psychedelics to catalyze profound changes in perception, emotion, and cognition that can contribute to transformative therapeutic experiences.

There is a substantial body of evidence suggesting the therapeutic potential of psychedelics, with various studies indicating their efficacy in treating mental health disorders. Interpersonal variability, however, is a critical factor that can modulate these outcomes, with elements such as set and setting, gut microbiota, and genetic factors playing a role in influencing the therapeutic efficacy of psychedelics.

Set and setting, referring to an individual's mindset and the environmental context, are critical aspects shaping the nature and quality of the psychedelic experience. This variability emphasizes the importance of a carefully prepared and supportive environment in clinical and therapeutic settings to maximize positive outcomes and minimize adverse reactions.

Regarding the gut–brain axis, emerging research has demonstrated that the gut microbiota can influence brain function and behavior, potentially modulating the response to psychedelics. This insight opens up new opportunities for microbiome-targeted interventions to optimize the therapeutic benefits of psychedelics, highlighting the role of dietary interventions, probiotics, prebiotics, and fecal microbiota transplantation.

Moreover, understanding the genetics of psychedelic response is vital for developing personalized medicine approaches in psychedelic therapy. Pharmacogenomic strategies involving genetic testing to predict an individual's drug response could potentially guide the selection of appropriate psychedelic substances and doses for each individual, thereby maximizing therapeutic and minimizing adverse effects.

Yet, while the therapeutic potential of psychedelics is exciting, it is crucial to be aware of this field's limitations and future directions. The challenges include the heterogeneity of the psychedelic experience, the lack of long-term studies, the current legal and regulatory hurdles, and the limited understanding of the precise role of gut microbiota. Future research should aim to overcome these limitations, particularly in developing personalized and safe psychedelic therapies, standardizing integration therapy, understanding the long-term effects, and exploring the potential of these substances for a broader range of disorders.

In conclusion, the psychedelic renaissance holds significant promise for the future of mental health treatment. To actualize the potential of psychedelic treatment, we need to undertake meticulous and extensive research to elucidate the intricate dynamics of the psychedelic experience. This requires delving into the profound shifts in consciousness triggered by these substances, recognizing the influence of individual variability in response to psychedelics, understanding the function of the gut–brain axis, and exploring the opportunities within personalized medicine. If navigated thoughtfully and responsibly, psychedelics could revolutionize mental health treatment, providing powerful new tools to promote healing and growth.

**Funding:** This research received no external funding.

**Institutional Review Board Statement:** Not applicable.

**Informed Consent Statement:** Not applicable.

**Data Availability Statement:** PubMed and SciFinder databases.

**Conflicts of Interest:** The author declares no conflict of interest.

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
