# Peer review of "Microbiome: The Next Frontier in Psychedelic Renaissance"

_jox, doi:10.3390/jox13030025_

Round 1
Reviewer 1 Report
The topic of the article - interaction between gut microbiome and psychedelics - definitely deserves academic attention and a good literature review.
I agree with the conclusions of the author about perspectives of the topic, also about needs for further research and changes in regulation. However, there are some issues with the article.
The most serious issue is that the figures in the article are directly copied from other publications. The references to their sources appear only as numbers in the text of the article. There are no references to the sources in captions of these figures and no indication of permission from the original authors. This minimal way of acknowledging the sources is probably not enough, it can be misleading and even bring legal trouble to the author or the journal. Re-publication of images usually needs explicit permissions from the original authors or copyright holders. See for example https://www.annualreviews.org/pb-assets/Authors%20Assets/FigurePermissionGuide_AR-1644519223260.pdf
For a literature review, it would be appropriate to report how the sources were found - e.g. searched databases, search keywords and time of the search, also the criteria for inclusion. This could increase transparency and convince the reader that the pool of available sources is well covered and the selection of the sources is objective and unbiased.
Technical error in a reference: row 488 "[14Error! Bookmark not defined.]"
I recommend publication of the article after the mentioned issues (especially the references for figures, but see also the comment about English) are solved.
English is generally very good, with some insignificant issues and typos that are not an obstacle to understanding the contents. E.g. in rows 71-72, please check the structure of the sentence: "Figure 1, Misera et al. illustrate the gut microbiota influences psychiatric treatment efficacy [2]." A good proofreading of the whole article would solve these minor points.
The style of the article is overflowing with academic eloquence, with the main ideas repeated multiple times in different wording. Occasionally, simpler expression would be more clear and true: e.g. in row 38: "symbiotic relationship between the gut microbiome and psychedelics". Here I would rather say "interaction" than "symbiotic relationship". Symbiosis usually refers to mutually beneficial relationship between two or more living organisms or ecosystems, and psychedelics are usually not considered as such.
Author Response
The most serious issue is that the figures in the article are directly copied from other publications. The references to their sources appear only as numbers in the text of the article. There are no references to the sources in captions of these figures and no indication of permission from the original authors. This minimal way of acknowledging the sources is probably not enough, it can be misleading and even bring legal trouble to the author or the journal. Re-publication of images usually needs explicit permissions from the original authors or copyright holders. See for example https://www.annualreviews.org/pb-assets/Authors%20Assets/FigurePermissionGuide_AR-1644519223260.pdf
Author Response: Thank you for your constructive feedback regarding the sourcing and permissions of the images used in the manuscript. I have now revised the manuscript to include appropriate references in the captions of the figures. In response to your feedback, where applicable, I have also sought and obtained the necessary permissions for using each image from the original authors or copyright holders. All images now carry the correct acknowledgment of their source, and any required copyright notices are displayed as per the terms of the respective permissions granted. Furthermore, I appreciate the resource you provided for seeking copyright permissions. It will undoubtedly be beneficial for future publications. I thank you again for your detailed review and valuable comments, which have greatly contributed to improving the quality of the manuscript.
For a literature review, it would be appropriate to report how the sources were found - e.g. searched databases, search keywords and time of the search, also the criteria for inclusion. This could increase transparency and convince the reader that the pool of available sources is well covered and the selection of the sources is objective and unbiased.
Author Response: I appreciate your insight regarding the transparency of the literature search process used in the review. To address your comment, I have revised the manuscript to more clearly detail the search strategy, including the databases searched, the keywords used, and the time frame of the search, which is covered in reference 6.
A technical error in a reference: row 488 "[14Error! Bookmark not defined.]"
Author Response: The technical error is rectified. Thank you.
I recommend publication of the article after the mentioned issues (especially the references for figures, but see also the comment about English) are solved.
Author Response: Thank you for your recommendations and your insightful, helpful, and thoughtful comments.
Comments on the Quality of English Language
English is generally very good, with some insignificant issues and typos that are not an obstacle to understanding the contents. E.g. in rows 71-72, please check the structure of the sentence: "Figure 1, Misera et al. illustrate the gut microbiota influences psychiatric treatment efficacy [2]." A good proofreading of the whole article would solve these minor points.
The style of the article is overflowing with academic eloquence, with the main ideas repeated multiple times in different wording. Occasionally, simpler expression would be more clear and true: e.g. in row 38: "symbiotic relationship between the gut microbiome and psychedelics". Here I would rather say "interaction" than "symbiotic relationship". Symbiosis usually refers to mutually beneficial relationship between two or more living organisms or ecosystems, and psychedelics are usually not considered as such.
Author Response: Thank you for your constructive feedback regarding the language and structure of the article. I appreciate your attention to detail, which undoubtedly enhances the clarity and readability of the article. To address your comments, I have employed a professional English proofreader to meticulously review the article and correct any minor typographical or language inconsistencies. The changes have been tracked in the revised and tracked changes documents for your convenience.
Regarding your point on using the term "symbiotic relationship", I acknowledge your suggestion and agree that "interaction" may be more accurate and straightforward in this context. This term has been replaced with "interaction" throughout the document to convey our intended meaning better.
Once again, I appreciate your valuable insights, which have helped improve our review's overall quality and clarity.
I appreciate your feedback regarding the style of the article and the perceived repetition of main ideas in different wording. I understand your concern and would like to clarify the approach. The topic is relatively new and complex and encompasses the interplay between the gut microbiome, psychedelics, mental health, and the brain. Given the interdisciplinary nature of this subject, I aimed to ensure that readers can follow the discussion and make connections between different concepts. By repeating some of the main ideas throughout the article, I intended to reinforce the key points and facilitate comprehension, particularly when discussing implications, personalized medicine, limitations, and future directions. My goal was to ensure that readers clearly understand the context and significance of these aspects, which often require referring back to previous discussions. Though, I have endeavored to minimize repetition in the revised version where necessary.
Reviewer 2 Report
The authors of the review entitled “Microbiome: The Next Frontier in Psychedelic Renaissance” address an interesting topic related to the high potential of human microbiota knowledge on promotes healthy status, exploring the intersection of the gut microbiome and potential psychedelic therapy. The work handles an interesting topic due to the importance of microbiota studies and the relevant participation of the microbiome in human health, however, the present work needs a few corrections before publication.
First, these main topics need revision:
There are several sentences without bibliographic support, considering that this work is a review. On the other hand, there are several sentences in plural with only one citation. In fact, the authors declare several times sentences with only one citation. That is critical for a review.
The authors discuss that microbiota plays an essential role in the psychedelic response, however, is needed more scientific support about this. There are few studies presented that look for a relationship between the effects of using psychedelics and microbiota composition. I miss a deep presentation about functional capacities based on metagenomic studies to support the ability to metabolize psychedelic compounds by microbiota microorganisms.
Specific corrections:
Line 42: correct the dot before citation: treatment. [4, 5, 6]
Line 45-46: Add citation to the sentence “The gut-brain axis is a complex bidirectional communication system…”. The same is in lines 50-51 about microbiome.
Line 57-58: The authors declare that microbiota effect modulating drugs has been well established, however, just add only one citation (11). Add more references.
Line 71: “Figure 1, Misera et al”, correct citation style. Which year? Also, in figure 1, indicate that the figure was taken or modified from (2). Same in all the other figures taken from other Works. Also, the figures look pixelled, in worse quality.
Line 112: It seems that CNS was not defined previously.
Line 314: Correct “LSD)”.
Author Response
There are several sentences without bibliographic support, considering that this work is a review. On the other hand, there are several sentences in plural with only one citation. In fact, the authors declare several times sentences with only one citation. That is critical for a review.
Author Response: Thank you for your valuable feedback and for pointing out the issue regarding the lack of bibliographic support in some sentences and the overuse of single citations in our review. I sincerely appreciate your thorough evaluation of the article. In response to your feedback, I have significantly increased the number of references from 60 to 97 to provide a more comprehensive and well-supported review. I have made a concerted effort to capture multiple citations for statements, as recommended by your suggestion.
I hope efforts to address your concerns have been successful and that you find the revised manuscript more satisfactory.
The authors discuss that microbiota plays an essential role in the psychedelic response, however, is needed more scientific support for this. Few studies are presented that look for a relationship between the effects of psychedelics and microbiota composition. I missed a deep presentation about functional capacities based on metagenomic studies to support the ability to metabolize psychedelic compounds by microbiota microorganisms.
Author Response: I appreciate your thoughtful evaluation of the manuscript and your critique regarding the lack of scientific support for the role of microbiota in the psychedelic response. As you rightly pointed out, limited studies are available that specifically explore the link between psychedelic usage and microbiota composition. The current body of scientific literature in this area is relatively sparse, making it challenging to provide an in-depth presentation on the functional capacities of microbiota microorganisms based on metagenomic studies. However, I believe that highlighting the existing studies and discussing the potential role of microbiota in the psychedelic response is crucial for drawing attention to this promising area of research. My intention in discussing this topic was to generate interest and inspire researchers to delve deeper into this fascinating area. Thank you once again for your valuable feedback. I greatly appreciate your insights and suggestions, which have undoubtedly improved the clarity and focus of our review.
Specific corrections:
Line 42: correct the dot before citation: treatment. [4, 5, 6] Author Response: Corrected, thank you.
Line 45-46: Add citation to the sentence “The gut-brain axis is a complex bidirectional communication system…”. The same is in lines 50-51 about microbiome.
Line 57-58: The authors declare that microbiota effect modulating drugs has been well established, however, just add only one citation (11). Add more references.
Author Response: Thank you again for pointing out the missing citations in the manuscript. Per your suggestion, I have added citations to the sentences in question. Furthermore, in response to your request for more references, I have significantly expanded the number of citations in the revised manuscript. The total references have been increased from 60 to 97, encompassing a broader range of relevant studies and enhancing the comprehensiveness of the review. Your expertise and insights are highly valued.
Line 71: “Figure 1, Misera et al”, correct citation style. Which year? Also, in figure 1, indicate that the figure was taken or modified from (2). Same in all the other figures taken from other Works. Also, the figures look pixelled, in worse quality.
Line 112: It seems that CNS was not defined previously.
Author Response: Regarding Figure 1, I apologize for the oversight in the citation style. I have made the necessary corrections and ensured that the proper citation style is used, including the reference number by Misera et al. I have also added the appropriate acknowledgment in the figure caption, indicating that the figure was taken or modified from the source, obtained the necessary copyright permissions from the publishers for all figures shown in the manuscript. Furthermore, I have taken steps to address the concern regarding the image quality of the figures. While I have made every effort to enhance the quality as much as possible within the constraints of the original papers, I understand the importance of presenting clear and legible figures. I have worked diligently to improve the figures' resolution and clarity, striving to provide the best representation of the original images.
I have revised the manuscript to include a clear definition of CNS before its use in line 112.
Line 314: Correct “LSD)”. Author Response: Corrected, ‘)’ removed. Thank you.
Reviewer 3 Report
In this review, Kargbo discussed how microbiota influences psychedelic effects. More specifically, the author particularly focused on how the gut microbiota modulate the effect psychedelic substances and how their interaction may help with personalized mental health treatment. This review has intensively discussed and commented each aspect of the points.
NA
Author Response
Author Response: Thank you for your favorable comment regarding the manuscript.
Round 2
Reviewer 1 Report
I see that the captions of reprinted figures now include specific references, I welcome the change.
It is good that the description of the literature search process has been added. I am not sure if the list of references is the best place for it, but better there than nowhere.
Language editing has changed some expressions for better. However, in row 223 describing Figure 4, "openness" has been replaced with "vulnerability" - while the figure itself clearly displays "openness", not "vulnerability". (Please check also the source of the figure.) Language editors should not change specific terms like that, but it happens. I advise the author to review all proposed changes with the content in mind and bravely reject any changes that may distort the meaning. Language editors specialize on language, not on the research topic, and critical review of the proposed changes is the responsibility of the author.
There seems to be a technical issue with numbering of the references in the list of references (double numbers, variations in format). This should be easy to correct for the final version.
The article has educational value, it summarizes a lot of previous work, and presents interesting hypotheses and directions for future research. While I do not have time to check all details, the references etc, I agree with the main conclusions and I think that the work is good enough to publish when the aforementioned minor issues get taken into account.
Author Response
Dear Reviewer,
Thank you for your astute observations and constructive feedback. You are correct in pointing out that the term 'openness' was inappropriately replaced with 'vulnerability' in row 223. This change resulted from language editing, and I appreciate you bringing this discrepancy to my attention.
I wholeheartedly agree with your advice on critically reviewing all proposed changes to ensure they do not distort the meaning of the original content. The specialists who edited the language are not experts in the research topic, and it is my responsibility as the author to scrutinize these modifications.
Moving forward, I have made the necessary corrections and ensured the terminology used in the manuscript aligns accurately with the scientific information presented. I sincerely appreciate your valuable input in maintaining the integrity of my work.
Thank you once again.
Best Regards,
Robert.